# Cytomegalovirus Infection after Allogeneic Hematopoietic Cell Transplantation under 100-Day Letermovir Prophylaxis: A Real-World 1-Year Follow-Up Study

**DOI:** 10.3390/v15091884

**Published:** 2023-09-06

**Authors:** Dukhee Nho, Raeseok Lee, Sung-Yeon Cho, Dong-Gun Lee, Eun-Jin Kim, Silvia Park, Sung-Eun Lee, Byung-Sik Cho, Yoo-Jin Kim, Seok Lee, Hee-Je Kim

**Affiliations:** 1Division of Infectious Diseases, Department of Internal Medicine, College of Medicine, The Catholic University of Korea, Seoul 06591, Republic of Korea; nhodh@catholic.ac.kr (D.N.); misozium03@catholic.ac.kr (R.L.); symonlee@catholic.ac.kr (D.-G.L.); 2Vaccine Bio Research Institute, College of Medicine, The Catholic University of Korea, Seoul 06591, Republic of Korea; 3Catholic Hematology Hospital, Seoul St. Mary’s Hospital, College of Medicine, The Catholic University of Korea, Seoul 06591, Republic of Korea; yeoni8383@gmail.com (E.-J.K.); silvia.park@catholic.ac.kr (S.P.); lee86@catholic.ac.kr (S.-E.L.); cbscho@catholic.ac.kr (B.-S.C.); yoojink@catholic.ac.kr (Y.-J.K.); leeseok@catholic.ac.kr (S.L.); cumckim@catholic.ac.kr (H.-J.K.)

**Keywords:** cytomegalovirus, prophylaxis, allogeneic hematopoietic cell transplantation, real-world data

## Abstract

The prevention and management of cytomegalovirus (CMV) reactivation is important to improve the outcomes of allogeneic hematopoietic cell transplantation (allo-HCT) recipients. The aim of this study was to analyze real-world data regarding the incidence and characteristics of CMV infections until 1 year after allo-HCT under 100-day letermovir prophylaxis. A single-center retrospective study was conducted between November 2020 and October 2021. During the study period, 358 patients underwent allo-HCT, 306 of whom received letermovir prophylaxis. Cumulative incidence of clinically significant CMV infection (CS-CMVi) was 11.4%, 31.7%, and 36.9% at 14 weeks, 24 weeks, and 1 year post-HCT, respectively. Through multivariate analysis, the risk of CS-CMVi increased with graft-versus-host disease (GVHD) ≥ grade 2 (adjusted odds ratio 3.640 [2.036–6.510]; *p* < 0.001). One-year non-relapse mortality was significantly higher in letermovir breakthrough CS-CMVi patients than those with subclinical CMV reactivation who continued receiving letermovir (*p* = 0.002). There were 18 (15.9%) refractory CMV infection cases in this study population. In summary, letermovir prophylaxis is effective at preventing CS-CMVi until day 100, which increased after the cessation of letermovir. GVHD is still a significant risk factor in the era of letermovir prophylaxis. Further research is needed to establish individualized management strategies, especially in patients with significant GVHD or letermovir breakthrough CS-CMVi.

## 1. Introduction

Human cytomegalovirus (CMV) is a globally prevalent virus that remains latent after primary infection, which can be reactivated in immunocompromised hosts, leading to end-organ disease. Despite prophylactic and pre-emptive therapy, CMV reactivation remains one of the most common and fatal complications of transplantation. Available antiviral agents of CMV were ganciclovir, valganciclovir, foscarnet, and cidofovir, all of which target *UL54* of the viral polymerase. Ganciclovir and valganciclovir are also known to involve viral kinase *UL97*. However, the limits of these antiviral agents, such as myelosuppression, nephrotoxicity, and cross resistance, pose challenges in transplantation [1]. Currently, prophylaxis emerges as a new strategy for CMV [2,3]. Letermovir, a recently introduced antiviral drug, inhibits pUL56, which is one of the core proteins of viral terminase complex involved in CMV DNA cleavage and packaging [4,5,6,7]. Unlike other antiviral agents, letermovir is specific for human CMV and has no significant antiviral activity against other herpes viruses [8].

Clinical trials have demonstrated the efficacy of letermovir prophylaxis in preventing CMV reactivation in allogeneic hematopoietic cell transplantation (allo-HCT) recipients [9]. Vyas et al. reported a systematic review and meta-analysis of real-world studies of letermovir [10]; however, a great part of those studies were conducted in US or Europe, and data from the Asia-Pacific region were scarce. The majority of the East Asian population is infected with CMV at young age; 94% of Korean adults are reportedly CMV seropositive, which contrasts with the results of the clinical trial conducted by Marty et al., where 60.8% of hematopoietic stem cell donors of the study population were CMV seropositive [9,11,12]. Differences in CMV risk factors, such as the type of transplantation, conditioning intensity, and CMV serostatus, might cause differences in the efficacy of letermovir prophylaxis. Additionally, follow-up observation until 1 year after transplantation showed a novel facet of CMV reactivation, such as breakthrough CMV reactivation during letermovir prophylaxis, CMV blips in the post-engraftment period, and late reactivation after the discontinuation of letermovir, which remain unexplored in the field of real-world clinical practice [13,14,15].

The aim of this study was to analyze real-world data for the incidence and characteristics of CMV infections until 1 year after allo-HCT under 100-day letermovir prophylaxis. We evaluated the proportion of patients with clinically significant CMV infection (CS-CMVi) and mortality at 14 weeks, 24 weeks, and 1 year after transplantation. We also analyzed risk factors of CS-CMVi, along with examining letermovir breakthrough CMV DNAemia, and the occurrence of CMV blips and refractory or resistant CMV infections.

## 2. Materials and Methods

### 2.1. Patients and Study Design

This retrospective single-institution cohort study was conducted at the Catholic Hematology Hospital, Seoul St. Mary’s Hospital. We reviewed the electronic medical records of patients over 18 years of age who underwent allo-HCT between 1 November 2020 and 31 October 2021. According to the approved indication, patients received letermovir prophylaxis if they were CMV seropositive and had undetectable plasma levels of CMV DNA up to 5 days before letermovir initiation. Patients on dialysis with creatinine clearance <10 mL/min and liver impairment equivalent to Child–Pugh class C were excluded from letermovir prophylaxis. Patients using rifampicin, rifabutin, nafcillin, and anticonvulsants, as well as patients simultaneously prescribed statins and cyclosporine, were excluded due to the risk of drug interactions with letermovir, according to data from previous clinical trials [2,9].

This study was approved by the Institutional Review Board of Seoul St. Mary’s Hospital (No. KC20MODP0823). Informed consent was not required due to the retrospective design of our study.

### 2.2. Institutional Protocol

Letermovir prophylaxis was initiated between day 0 and day 28 after allo-HCT. Plasma CMV DNA polymerase chain reaction (PCR) was performed biweekly during hospitalization and at every visit after discharge. Letermovir prophylaxis was continued until day 100 after allo-HCT, unless the patient showed CS-CMVi, such as CMV DNAemia requiring pre-emptive therapy or CMV end-organ disease. Letermovir prophylaxis was discontinued if the underlying hematological disease relapsed. We categorized the patients according to their risk of CMV reactivation. The high-risk group included patients with related donors with at least one mismatch at one of the three HLA gene loci (HLA-A, B, or HLA-DR), those with unrelated donors with at least one mismatch at one of the four HLA loci (HLA-A, B, C, or DRB1), patients with haploidentical transplants, cord blood transplants, and the presence of clinically significant graft-versus-host disease (GVHD) over grade 2 that led to the use of steroids with a dose equivalent to at least 20 mg of prednisolone per day for more than 2 weeks at the time of letermovir initiation. Patients who did not meet the above criteria were classified as low risk. Patients received 480 mg of letermovir daily, but those prescribed cyclosporine received 240 mg of concomitant letermovir owing to drug interactions. CS-CMVi was defined as plasma CMV DNA level >500 IU/mL in high-risk patients and >1000 IU/mL in low-risk patients until day 100 after transplantation. In these cases, letermovir prophylaxis was stopped and pre-emptive therapy started. After day 100, pre-emptive treatment was initiated if the plasma CMV DNA level was >1000 IU/mL, regardless of the risk group.

### 2.3. Definitions

CMV viral load was measured via real-time quantitative PCR test using artus CMV QS-RGQ MDx Kit (QIAGEN, Hilden, Germany) and Rotor-Gene Q (QIAGEN) based on the manufacturer’s instructions. The lower limit of quantification of the assay was 69.7 IU/mL. PCR results were classified into three categories: ‘Not detected,’ when no plasma CMV DNA was detected; ‘result under 69.7 IU/mL’, when CMV DNA was detected but was not quantifiable; and CMV DNA titer reported as numeric value in IU/mL, when CMV DNA was detected and quantifiable. In this study, quantifiable CMV DNA reported as a numeric value was defined as CMV reactivation. The definition of CMV end-organ disease was adopted from a previous description of Ljungman et al. [16]. An increasing or persistent CMV load after at least 2 weeks of appropriate treatment was defined as a refractory or probable refractory CMV infection, respectively [17]. CMV blip was interpreted as the presence of CMV DNAemia at any level in a single plasma specimen preceded and succeeded by an undetectable PCR specimen drawn 7 days apart [13,14].

### 2.4. Statistical Analysis

All analyses were performed using SPSS Statistics software, version 24.0 (IBM Corp., Armonk, NY, USA). Chi-square analysis or Fisher’s exact test was used to compare categorical variables, and a Student’s *t*-test or Mann–Whitney U test was used to compare continuous variables. The time to CS-CMVi between the groups was compared using the Kaplan–Meier curve and the Log-rank test. Multivariate logistic regression analysis was also performed. A *p* value < 0.05 was considered to be statistically significant.

## 3. Results

### 3.1. Study Population and Baseline Characteristics

During the study period, 358 patients underwent allo-HCT, 306 of whom received letermovir prophylaxis (Figure 1). Fifty-two patients were excluded from analysis, mainly due to the detection of plasma CMV DNA (n = 12), CMV seronegativity (n = 7), the use of contraindicated drugs (n = 7), and other situations in which letermovir was not prescribed to the patients (i.e., physician preference, n = 26). Less than 2% of allo-HCT recipients (n = 7) were CMV seronegative at the time of transplantation. Two patients underwent transplantation twice during this period, and they were counted for each episode. Of the 306 patients who received letermovir prophylaxis, 47.4% were defined as high risk, while 52.6% were at low risk of CMV reactivation (Figure 1).

Table 1 shows the baseline characteristics of patients. The median age of the patients was 50 years (range, 18–73), and 51.0% were men. Acute myeloid leukemia was the most common underlying disease, followed by acute lymphocytic leukemia, myelodysplastic syndrome, myeloproliferative neoplasms, and multiple myeloma. Other diseases included lymphoma (n = 8), chronic myelomonocytic leukemia (n = 4), and mixed phenotype acute leukemia (n = 2). A total of 165 patients underwent myeloablative conditioning. Patients with matched sibling donors and matched unrelated donors accounted for 27.5% and 38.2%, respectively. Family mismatched transplantation was performed in 26.1% of patients, while double-cord transplantation was performed in 8.2% of patients. The majority of the donors were CMV seropositive (79.1%) (Table 1). The distribution of the day of letermovir initiation after transplantation is shown in Figure 2. Patients began letermovir prophylaxis at a median date of 2 days (range, day 0–27) after transplantation.

### 3.2. Discontinuation of Letermovir

Seventy-four patients (24.2%) discontinued letermovir before day 100 after transplantation, mainly because of letermovir breakthrough CS-CMVi (n = 35, 11.4%), the relapse of underlying hematologic disease (n = 19, 6.2%), death (n = 9, 2.9%), loss to follow-up (n = 3, 1.0%), the initiation of continuous renal replacement therapy (n = 1, 0.3%), the adverse effects of letermovir, such as nausea/vomiting (n = 1, 0.3%), and poor physical condition making them incapable of oral intake (n = 6, 2.0%). Intravenous letermovir has been administered since March 2021, allowing a larger number of patients to continue receiving prophylaxis during the study period.

### 3.3. CMV Reactivation and Mortality

Clinically significant CMV infection was observed in 11.4% (n = 35) of patients at 14 weeks, 31.7% (n = 97) of patients at 24 weeks, and 36.9% (n = 113) of patients at 1 year after allo-HCT. Any level of CMV reactivation, which included both CS-CMVi and CMV DNAemia that did not require pre-emptive therapy, occurred in 26.5% (n = 81), 64.1% (n = 196), and 73.9% (n = 226) of patients at 14 weeks, 24 weeks, and 1 year after transplantation, respectively. All-cause mortality was 7.2% (n = 22), 10.5% (n = 35), and 21.6% (n = 66) at 14 weeks, 24 weeks, and 1 year, respectively (Table 2). CMV blip was observed in 11.4% of patients (n = 35/306); it was observed in 33 patients at 14 weeks and 35 patients at 24 weeks after transplantation.

We compared the incidence of CS-CMVi in high- and low-risk groups 1 year post-transplantation. CS-CMVi occurred 15.2% (n = 22/145) and 8.1% (n = 13/161) of patients at 14 weeks, 35.2% (n = 51/145) and 28.6% (n = 46/161) of patients at 24 weeks, and 39.3% (n = 57/145) and 34.9% (n = 56/161) of patients at 1 year after transplantation in the high- and low-risk groups, respectively. The cumulative incidence of CS-CMVi was significantly higher in the high-risk group than in the low-risk group at 14 weeks (*p* = 0.043). Compared to the low-risk group, the high-risk group showed a consistently higher incidence of CS-CMVi at 24 weeks and 1 year after transplantation; however, the cumulative incidence rate was not statistically significant (*p* = 0.090 at 24 weeks, *p* = 0.156) at 1 year. Notably, a sharp increase in CS-CMVi was observed from day 130 to 180 after transplantation (Figure 3).

We investigated the incidence of CS-CMVi according to the time of initiation of letermovir prophylaxis. Patients were divided into early- and late-initiation groups with a cut-off of day 2 post-allo-HCT based on the distribution of our institution (Figure 2). The early-initiation group included patients who began letermovir on days 0–2 after allo-HCT (n = 156), whereas the late initiation group included patients who began letermovir on days 3–28 after transplantation (n = 150). The cumulative incidence of CS-CMVi between the early- and late-letermovir initiation groups was not significantly different up to 1 year after transplantation (*p* = 0.746 at 14 weeks, *p* = 0.641 at 24 weeks, and *p* = 0.925 at 1 year after transplantation).

### 3.4. Risk Factors of Clinically Significant CMV Infection

We analyzed factors affecting CMV reactivation, such as the type of conditioning, underlying malignancies, the type of transplantation, the CMV serostatus of the donor, and the presence of GVHD before CS-CMVi. Following a multivariate analysis, only GVHD ≥ grade 2 was deemed to be a significant risk factor of CMV reactivation at 1 year after transplantation (adjusted OR 3.64, 95% CI [2.036–6.510]; *p* < 0.001). Myeloablative conditioning; lymphoid lineage malignancies such as acute lymphocytic leukemia, lymphoma, and multiple myeloma; matched sibling donor transplantation; and CMV seronegative donors did not significantly influence the risk of CMV reactivation (Table 3).

### 3.5. Letermovir Breakthrough CMV Reactivation

We compared the 1-year relapse-free survival of patients based on their CMV reactivation status at 14 weeks after transplantation (Figure 4). After excluding 55 relapsed patients, 190 patients did not show any level of CMV reactivation (group A), but letermovir breakthrough CMV reactivation occurred in 61 patients. Among them, 35 patients had CMV DNAemia that did not require pre-emptive therapy and completed letermovir prophylaxis (group B), whereas 26 patients had CS-CMVi and stopped letermovir in favor of the initiation of pre-emptive therapy (group C). The 1-year relapse-free survival was significantly higher in groups A and B, which continued letermovir, than in group C, which discontinued letermovir in favor of pre-emptive therapy (group A vs. C, *p* = 0.001; group B vs. C, *p* = 0.002) (Figure 4).

### 3.6. Refractory/Probable Refractory CMV Infection

Among the patients who developed CS-CMVi, we identified 15.9% (n = 18) of refractory/probable refractory CMV infections. Among them, seven cases occurred during letermovir prophylaxis at a median of day 16 post-HCT; other cases happened under ganciclovir/valganciclovir pre-emptive therapy after the cessation of letermovir at day 100 post-HCT. Our institution performs CMV drug resistance testing via the DNA sequencing of genes *UL54* and *UL97*. Methods of PCR, detection, and the interpretation of variants are described by Chae et al. [18]. We performed DNA sequencing of 13 patients with refractory/probable refractory CMV infection, and 7 patients appeared to have mutations in *UL54* and *UL97*. Among these seven patients with mutations, three and two patients had variants of unknown significance of *UL54* and *UL97* genes, respectively, one patient had the *UL97* drug resistance mutation A594V, and one patient had the deletion of the *UL97* gene. All of these refractory/probable refractory cases experienced significant GVHD that happened at a median of day 29 post-HCT; GVHD occurred before CMV infection in 13 cases (range of days from GVHD to CMV infection; 0–444), whereas 5 cases happened after CMV infection (range of days from CMV infection to GVHD; 1–51).

## 4. Discussion

In this study, we observed that letermovir prophylaxis was effective at reducing CS-CMVi until day 100, but incidence of CS-CMVi sharply increased from day 130 to 180 after transplantation. Moreover, mortality was significantly higher in cases of CS-CMVi, unlike subclinical CMV reactivation. Cumulative incidence of CS-CMVi was 11.4%, 31.7%, and 36.9% at 14 weeks, 24 weeks, and 1 year after transplantation, respectively. Incidence of CMV reactivation and CMV diseases declined compared to previous studies conducted at our institution. For instance, before the introduction of letermovir, 56.1% of allo-HCT patients developed any level of CMV reactivation between engraftment and day 100 after transplantation, compared to 26.5% treated with letermovir prophylaxis [19,20]. Our incidence of CS-CMVi is higher than that of Marty et al., who reported CS-CMVi incidence rates of 7.7% and 17.5% at 14 weeks and 24 weeks after transplantation, respectively [9]. This difference could be explained based on higher proportions of CMV-seropositive donors (79.1% vs. 61.7%), the use of antithymocyte globulin (79.4% vs. 37.5%), and the presence of high-risk patients (47.4% vs. 32.4%) in our institution. The assessment of CMV blip in an outpatient setting was difficult since the definition adopted in this study was identifying two undetected CMV levels 7 days apart before and after a positive CMV titer. We conceived a category called CMV DNAemia that did not require pre-emptive therapy, which constituted a quantifiable CMV titer in plasma, but did not reach the cut-off value for pre-emptive therapy. It included CMV blip and persistent low-titer CMV reactivation (Table 2).

There is no controversy regarding the effectiveness of letermovir prophylaxis until day 100 after allo-HCT; however, an increased incidence of CS-CMVi, especially at 4–6 months after transplantation, has been observed. In our study, we examined whether traditional risk factors of CMV have been changed under letermovir prophylaxis [21]. In contrast to other risk factors that did not demonstrate statistical significance, GVHD emerged as the sole risk factor, increasing the risk of CS-CMVi 3.6-fold. In such cases, prolonged letermovir prophylaxis may help in terms of delaying CMV reactivation until the restoration of cellular immunity. Studies of extended letermovir prophylaxis are in progress, and its effect on the prevention of CMV infections in specific risk group needs further investigation [22].

Data regarding CMV resistance is important but still lacking. Our institution performs resistance tests targeting *UL54* and *UL97* genes [18,23,24]. Cases of refractory/probable refractory CMV infection were 15.9% in this study, and 1.8% (n = 2) had known CMV resistance mutations. Variants of unknown significance in *UL54* or *UL97* mutation made up 4.4% (n = 5) of all variants. Although we did not perform sequencing for the *UL51*, *UL56*, or *UL89* genes, which are associated with letermovir drug resistance [25,26], as a routine process during the study period, the sequencing of these genes is also necessary to identify CMV drug resistance in this era of letermovir prophylaxis.

We presumed that the early initiation of letermovir would lead to better control and prevention of CS-CMVi. A previous study reported a significant difference in CMV DNAemia between the early (day 0–1) and late (day 2–27) prophylactic groups [27]. Compared to other studies, letermovir prophylaxis was started relatively early in our institution [10], recording a positively skewed distribution with a median of day 2 (interquartile range: day 2–4) over a period of 28 days (Figure 2). The cumulative incidence of CS-CMVi between the early (day 0–2) and late (day 3–28) letermovir initiation groups at our institution was not significantly different up to 1 year after transplantation. A possible explanation may be the tendency toward early commencement of prophylaxis. Whether the early initiation of letermovir prophylaxis influences the incidence of CS-CMVi should be checked using data with an even distribution of initiation days.

We closely examined the first 14 weeks after transplantation, which was the period of letermovir prophylaxis. Mortality was significantly higher in patients with letermovir breakthrough CS-CMVi. This result could be explained based on the severity of those patients. Among the 26 patients with letermovir breakthrough CS-CMVi, 10 died in the first year after transplantation due to septic shock, fungal infection, GVHD, and engraftment failure. Moreover, seven patients were identified as having refractory/probable refractory CS-CMVi. However, not all cases of letermovir breakthrough CMV reactivation had a poor prognosis. In subjects with letermovir breakthrough CMV reactivation that did not require pre-emptive therapy, relapse-free survival 1 year after transplantation was not different from that of patients with no CMV reactivation. In this patient group, we observed either persistent but self-recovering low-titer CMV reactivation or a single episode of CMV reactivation disappeared at the next examination without any intervention. This low-titer CMV reactivation was under the defined cut-off value for pre-emptive therapy, i.e., 500 or 1000 IU/mL for high- and low-risk groups, respectively. Regardless of the interval between specimens or the peak viral load under the cut-off value, single episodes of CMV reactivation were frequently observed and did not appear to be related to the non-relapse mortality. Such episodes of CMV reactivation can also be explained based on the mechanism of action of letermovir. Letermovir inhibits the CMV DNA terminase complex, which cuts viral DNA into viral genomes to be packaged into mature viral particles [4,5,6,7]. Cassaniti et al. detected the accumulation of non-replicative CMV DNA called ‘abortive CMV DNA’, which may mimic CMV reactivation [27]. Laboratory techniques could help to distinguish between real CMV reactivation leading to CS-CMVi and abortive subclinical CMV DNA particles; however, immediate differentiation in real-world clinical practice is difficult. We used a cut-off value of plasma CMV titer for CS-CMVi and started pre-emptive therapy; the adjustment of this cut-off value to reduce the detection of subclinical CMV infection and select CS-CMVi requires further research.

The strength of our study is that we provided real-world data regarding letermovir prophylaxis in a population with high CMV seroprevalence. We categorized letermovir breakthrough CMV reactivation and compared mortality rates. We noted that not all CMV reactivation had poor prognosis, and some reactivation could be observed without stopping letermovir. Our study has limitations, including its single-center retrospective design. We identified refractory/probable refractory CMV reactivation that occurred during letermovir prophylaxis, but the *UL56* gene mutation was not routinely analyzed in our institution during the study period. Further research is needed regarding *UL56* mutations and their clinical implications. Moreover, cellular immunity, along with CMV reactivation, should be investigated from clinical sample per period after transplantation.

In summary, letermovir prophylaxis led to a lower incidence of CS-CMVi than previously reported. However, we found that CS-CMVi increased in patients with GVHD and at 4–6 months after transplantation. During letermovir prophylaxis, not all patients with letermovir breakthrough CMV reactivation had a poor prognosis. Further study is needed to find management strategies that more effectively prevent CMV infection, such as prophylaxis in especially high-risk patients.

## Figures and Tables

**Figure 1 viruses-15-01884-f001:**
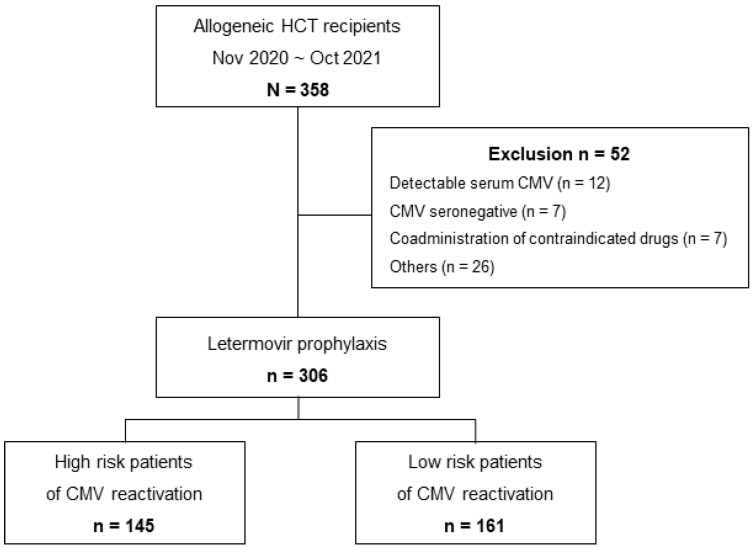
Flow chart of study population and CMV reactivation risk stratification.

**Figure 2 viruses-15-01884-f002:**
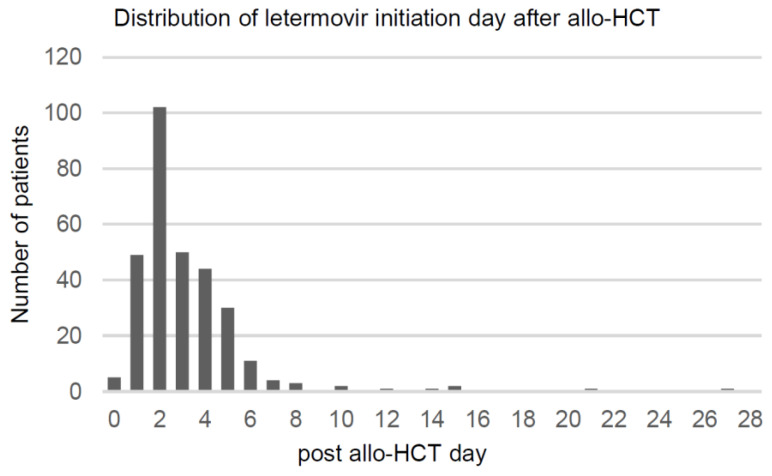
Distribution of the day of letermovir initiation after transplantation.

**Figure 3 viruses-15-01884-f003:**
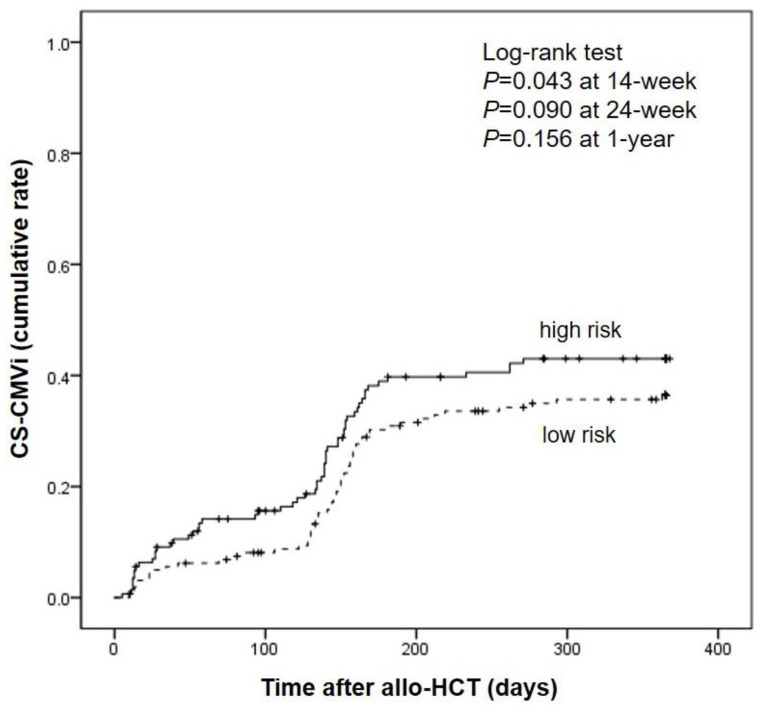
Cumulative incidence rate of clinically significant CMV infection. Cumulative incidence rate of clinically significant CMV infection (CS-CMVi) in a time-to-event analysis up to 1 year after allogeneic hematopoietic cell transplantation (allo-HCT), with data stratified according to the risk group.

**Figure 4 viruses-15-01884-f004:**
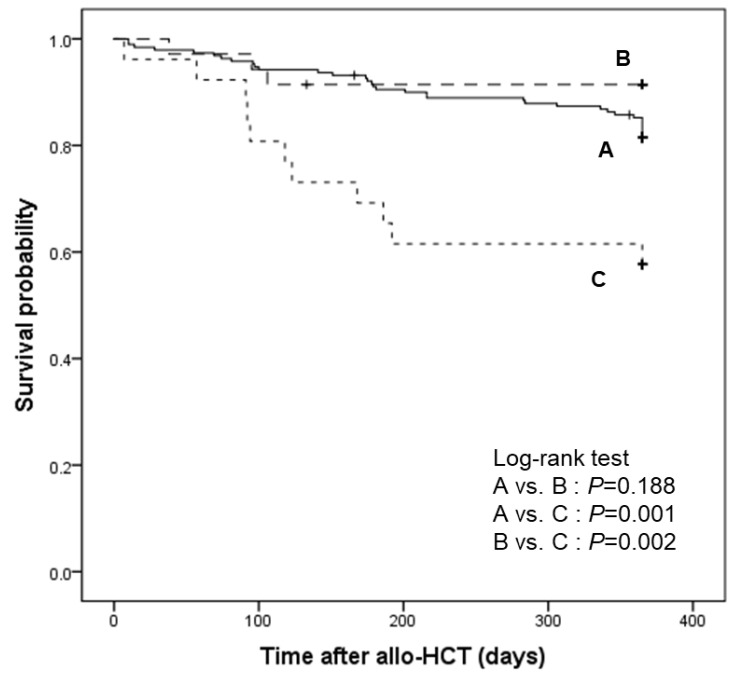
One-year relapse-free survival of patients according to CMV reactivation status at 14 weeks after transplantation. Patients without CMV reactivation (group A, n = 190), patients with letermovir breakthrough CMV DNAemia who did not require pre-emptive therapy and continued letermovir prophylaxis (group B, n = 35), and patients with letermovir breakthrough CMV DNAemia requiring pre-emptive therapy, that is, clinically significant CMV infection (CS-CMVi) (group C, n = 26) at 14 weeks after transplantation.

**Table 1 viruses-15-01884-t001:** Baseline characteristics of patients.

	(N = 306), n (%)
Age (years), median (range)	50 (18–73)
Male gender	156 (51.0%)
Underlying disease	
Acute myeloid leukemia	133 (43.5%)
Acute lymphocytic leukemia	72 (23.5%)
Myelodysplastic syndrome	53 (17.3%)
Myeloproliferative neoplasm	29 (9.5%)
Multiple myeloma	5 (1.6%)
Other diseases ^1^	14 (4.6%)
Type of conditioning regimen	
Myeloablative conditioning	165 (53.9%)
Non-myeloablative conditioning	141 (46.1%)
Use of antithymocyte globulin	243 (79.4%)
Type of donor	
Matched sibling donor	84 (27.5%)
Matched unrelated donor	117 (38.2%)
Family mismatched transplantation	80 (26.1%)
Double cord blood transplantation	25 (8.2%)
Donor/Recipient CMV serostatus	
D+/R+	242 (79.1%)
D−/R+	39 (12.7%)
Unknown	25 (8.2%)
Risk stratification for CMV reactivation	
High risk	145 (47.4%)
Low risk	161 (52.6%)
Start day of letermovir after HCT (days), median (range)	2 (0–27)
Acute GVHD of grade ≥ 2 at initiation of letermovir	1 (0.3%)

^1^ Other diseases included lymphoma (n = 8), chronic myelomonocytic leukemia (n = 4), and mixed phenotype acute leukemia (n = 2).

**Table 2 viruses-15-01884-t002:** CMV reactivation and mortality at 14 weeks, 24 weeks and 1 year after allogeneic hematopoietic cell transplantation.

	(N = 306), n (%)		
	14 Weeks	24 Weeks	1 Year
Any level of CMV reactivation	81 (26.5%)	196 (64.1%)	226 (73.9%)
CS-CMVi, received pre-emptive therapy	35 (11.4%)	97 (31.7%)	113 (36.9%)
CMV end-organ disease	8 (2.6%)	20 (6.5%)	25 (8.2%)
CMV DNAemia, not requiring pre-emptive therapy	46 (15.0%)	99 (32.4%)	113 (36.9%)
All-cause mortality	22 (7.2%)	32 (10.5%)	66 (21.6%)
CMV-related mortality	1 (0.3%)	3 (1.0%)	5 (1.6%)
Follow-up loss	2 (0.3%)	4 (1.3%)	9 (2.9%)

**Table 3 viruses-15-01884-t003:** Multivariate logistic regression analysis of risk factors of clinically significant CMV infection.

Variable	OR	95% CI	*p* Value
Myeloablative conditioning	1.054	0.648–1.712	0.833
Lymphoid lineage malignancies	1.355	0.794–2.314	0.266
Matched sibling donor HCT	0.937	0.542–1.618	0.815
CMV seronegative donor	0.845	0.402–1.778	0.657
GVHD (≥grade 2)	3.64	2.036–6.510	<0.001

Abbreviations. CMV, cytomegalovirus; OR, odds ratio; CI, confidence interval; HCT, hematopoietic cell transplantation; GVHD, graft versus host disease.

## Data Availability

Original data are available from the corresponding author upon request via email.

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
