# Peer review of "Cytomegalovirus Infection after Allogeneic Hematopoietic Cell Transplantation under 100-Day Letermovir Prophylaxis: A Real-World 1-Year Follow-Up Study"

_viruses, 2023, doi:10.3390/v15091884_

Round 1
Reviewer 1 Report
In this study the authors analysed the incidence and consequences of clinically significant HCMV infections after allo-HCT, all of which received Letermovir prophylaxis.
This study is well done and contributes to the knowledge of the effectiveness of Letermovir prophylaxis in adult patients after all-HCT. In this study 306 patients with Letermovir prophylaxis were included, the donor seroprevalence rate was 79% and the patients were followed-up for one year. The authors showed that Letermovir breakthrough CS-CMVi (26/251) were significantly associated with a lower one-year relapse free survival of patients, that the incidence of CS-CMVi increased after cessation of Letermovir prophylaxis, and GVHD was a risk factor for CS-CMVi. Please find below my comments.
Comments:
Materials and Methods:
2.2. Why do patients with haploidentical transplants belong to the high risk group?
2.3. CMV blips were described in this section but it was never mentioned in the results part. Did CMV blips occur in this study cohort?
Did you use serum or plasma for CMV DNA level determination? It is not concisely used throughout the manuscript.
Results:
3.3. Please explain why you discriminate between early and late initiation of Letermovir prophylaxis and why you have chosen 0-2 days for early and > 2 days for late. Of note: in the discussion part early was 0-1 days and late was 2-27 days!
3.6.
- Please clarify whether the refractory/probable refractory CMV infection was seen under ganciclovir or letermovir medication. If a refractory CMV infection is seen under Letermovir medication it might be important to test for potential resistance mutations at least in UL56.
- It would be interesting to describe the variants of unknown significance as well as the known resistance mutation and deletion in UL97.
- Please include in the Materials and Methods section how the Ganc resistance testing has been performed.
- Please include the range of days how long before and after the CMV infection the patients experienced a GVHD.
Reviewer 2 Report
The manuscript is a timely description of a current topic, it is important and will doubtless be of great interest to many readers. I have no major points of criticism, and enjoyed reading the manuscript. As minor points: I would welcome a paragraph concerning the limitations of the study in the discussion. The English could also be improved, but is undertandable.
The English should be checked as there are some imperfections. However this is not at a level to prevent understanding of the paper.
Author Response
Thank you for taking the time to review this manuscript. Please, find the detailed responses below and the corresponding revisions and corrections highlighted in the re-submitted files.
Point-by-point response to Comments
Comment 1. I would welcome a paragraph concerning the limitations of the study in the discussion.
Response 1. Thank you for pointing this out. We have updated limitations of our study in the discussion section (page 10/12, line 335-340), highlighted. Below are the text added in the manuscript.
“Our study has limitations, including its single-center retrospective design. We identified refractory/probable refractory CMV reactivation that occurred during the use of letermovir, but UL56 gene mutation was not examined routinely during this study. Further research is needed on UL56 mutations and clinical implications. Moreover cellular immunity along with CMV reactivation should be investigate from clinical sample per period after transplantation.”
Reviewer 3 Report
HCMV can cause serious diseases in immunocompromised patients. Current antiviral inhibitors (ganciclovir, cidofovir and foscarnet) all target the viral DNA polymerase. They have adverse effects and prolonged treatment can select for drug resistance mutations. Thus, we need new drugs. In this context, Guo and co- analyze real-world data for incidence and characteristics of CMV infections until 1-year after allo-HCT under 100-day letermovir prophylaxis
This study is highly relevant to the field since we need new drugs targeting other stages of the viral replication.
Comments:
1. In the introduction - What about letermovir mechanism of action? The authors should extend the DNA packaging steps and cite Heming et al., 2017 and Ligat et al., 2018.
The manuscript is well written.
Author Response
Thank you for taking the time to review this manuscript. Please, find the detailed responses below and the corresponding revisions and corrections highlighted in the re-submitted files.
|
2. Questions for General Evaluation |
Reviewer’s Evaluation |
Response and Revisions |
|
Does the introduction provide sufficient background and include all relevant references? |
Can be improved |
We added references that you recommended for letermovir MoA. See below the point-by-point response for detail. |
|
Are all the cited references relevant to the research? |
Can be improved |
We revised our references and saved the most relevant articles. (For example, article about CMV and solid organ transplantation was excluded) |
Point-by-point response to Comments
Comment 1. In the introduction - What about letermovir mechanism of action? The authors should extend the DNA packaging steps and cite Heming et al., 2017 and Ligat et al., 2018.
Response 1. Thank you for pointing this out. We elaborate refractory CMV infection in the Results section with drug resistance test and DNA sequencing. It would be helpful for readers to mention letermovir mechanism of action at DNA level. This additional explanation can be found at the Introduction section (page 1-2, line 40-49), highlighted, along with the references you recommended.